# Small RNA sequencing analysis of exosomes derived from umbilical plasma in IUGR lambs

Jiawei Lu [1], Huixia Li [1,3✉], Xiaomin Zheng [2,3✉], Yuan Liu[1] & Peng Zhao[1]

During the summer, pregnant ewes experience heat stress, leading to the occurrence of IUGR lambs. This study aims to explore the biomarkers of exosomal miRNAs derived from umbilical plasma in both IUGR and normal Hu lambs. We establish a heat-stressed Hu sheep model during mid-late gestation and selected IUGR and normal lambs for analysis. Exosomes from umbilical plasma were separated and small RNA sequencing is used to identify differentially expressed miRNAs. Next, we utilize MiRanda to predict the target genes of the differentially expressed miRNAs. To further understand the biological significance of these miRNAs, we conduct GO and KEGG pathway enrichment analysis for their target genes. The study's findings indicate that oar-miR-411a-5p is significantly downregulated in exosomes derived from umbilical plasma of IUGR lambs, while oar-miR-200c is significantly upregulated in the HS-IUGR group ($P < 0.05$). Furthermore, GO and KEGG enrichment analysis demonstrate that the target genes are involved in the Wnt, TGF-beta, and Rap1 signaling pathways. miRNAs found in exosomes have the potential to be utilized as biomarkers for both the diagnosis and treatment of IUGR fetuses.

[1] College of Animal Science and Technology, Nanjing Agricultural University, 210095 Nanjing, China. [2] Research Institute for Reproductive Medicine and Genetic Diseases, Wuxi Maternity and Child Health Hospital, Wuxi 214002 Jiangsu, China. [3]These authors jointly supervised this work: Huixia Li, Xiaomin Zheng. ✉email: lihuixia@njau.edu.cn; 972514523@qq.com

**H**eat stress has a significant impact on both human and animal production, including growth, development, and reproduction[1]. There are many factors that cause intrauterine growth retardation (IUGR), such as maternal malnutrition, placental dysfunction, and fetal factors, genetic factors also account for a major proportion of the occurrence of IUGR[2,3]. In particular, sheep are vulnerable to heat stress when the temperature rises above 25 °C, which challenges their natural temperature regulation mechanisms and can lead to reduced performance and health issues[4]. As global warming continues, sheep will face even more intense and prolonged heat stress[5]. As a result, heat stress is a major factor that affects the development of the sheep industry. Exposure of sheep to heat stress has increasingly severe consequences due to the rising temperatures associated with global climate change[6]. When pregnant sheep are exposed to heat stress, it negatively impacts fetal growth and the weight of newborn lambs[6]. Heat stress begins to affect the fetus during mid-gestation, and by late gestation, the size of the fetus is less than two standard deviations from that of a normal sheep[7]. During late gestation and in newborn lambs from heat-stressed ewes, there is a change in fetal morphology that the average fetal weight and liver weight decrease, which suggests the presence of IUGR[8].

This IUGR is caused by the elevated ambient temperature experienced by the ewes during the pregnancy period, and there is a strong correlation between the maximum temperature of the uterus during pregnancy and the birth weight of the offspring[9,10]. IUGR is a condition where the weight of the fetus is less than two standard deviations of the average weight for the same age, or below the 10th percentile of the normal weight for the same age, and IUGR is one of the significant complications that can arise during pregnancy[11]. Lambs born under severe natural conditions, such as cold and heat stress, experience a decrease in birth weight[12]. The impact of ambient temperature on pregnancy varies depending on the stage of gestation, with late pregnancy being more vulnerable than early pregnancy, and there is an almost linear correlation between the ewe's thermal peak and the birth weight of their offspring[10].

Exosomes, which are membranous vesicles with a diameter of 30–150 nm, are produced and secreted by living cells[13]. They have a round or cup-like shape that can be observed through electron microscopy[14]. Exosomes are small vesicles that play a crucial role in intercellular communication, and there are around $10^{14}$ exosomes in the human body, with each cell producing an average of 1000–10,000 exosomes[15]. These vesicles can be found in almost all tissues, intercellular spaces, and body fluids, such as blood, saliva, urine, and breast milk, and they carry various molecules, including proteins, miRNAs, lncRNAs, circRNAs, and mRNA, which are involved in intracellular signal transduction and help regulate the biological processes of cells[16]. Exosomes can also be produced by the placenta and have the potential to serve as biomarkers for pregnancy diagnosis by carrying miRNA into the maternal circulation[17]. In addition, exosomes secreted by the endometrium enhance the embryo's adhesion ability[18]. As gestation progresses, the concentration of serum exosomes from the gestational day of 90 pregnant sheep also increases[19]. Therefore, exosomes play a critical role in both pregnancy diagnosis and embryo implantation.

To establish a model of heat stress during mid-late gestation, Hu sheep were used as experimental animals in this study. After the gestation period, IUGR and healthy lambs were distinguished. Then exosomes from the umbilical plasma of both IUGR and healthy lambs were separated. Small RNA sequencing was conducted to identify the small RNAs expressed in exosomes and to understand their roles in the development of IUGR due to heat stress. Our findings demonstrated that miRNAs present in exosomes from umbilical plasma have the potential to serve as biomarkers for both the diagnosis and treatment of IUGR in fetuses.

## Results

**Establishment of the heat-stressed Hu sheep model during mid-late gestation**. The ambient temperature of the sheep shed fluctuated between 5 and 25 °C in Jan, Feb, and Mar, and undulated from 20 to 40 °C in Jun, Jul, and Aug (Fig. 1a). The effective temperature of pregnant Hu sheep was similar to the ambient temperature (Fig. 1b). The temperature-humidity index (THI) showed most pregnant Hu sheep were in a no-heat-stressed state in Jan, Feb, and Mar, and only a few days suffered mild stress, nevertheless, most pregnant Hu sheep underwent high and even severe stress in Jun, Jul, and Aug (Fig. 1c). The rectal temperature and respiration rate of pregnant Hu sheep in Jun, Jul, and Aug were higher than those in Jan, Feb, and Mar (Fig. 1d, e).

**Body measurement traits of normal and IUGR lambs**. The IUGR lambs exhibited significantly lower weight, height, chest depth, chest width, chest circumference, and circumference of the forearm bone compared to the normal lambs ($P < 0.01$, Fig. 2). However, there was no significant difference in body length between the two groups ($P > 0.05$, Fig. 2b). These findings suggest that heat stress during pregnancy had a negative impact on the body measurement traits of the IUGR lambs.

**Isolation and identification of exosomes from umbilical plasma**. Transmission electron microscopy analysis revealed the presence of a significant number of "cup and plate" exosomes in the umbilical plasma of both IUGR and normal lambs (Fig. 3a). The mean size of the exosomes was 139.95 nm in the CON group and 130.95 nm in the HS-IUGR group, as shown in Supplementary Table 1. Nanoparticle tracking analysis indicated that exosomes with a diameter of 138.2 nm accounted for 99.3% of the CON group, while exosomes with a diameter of 126.9 nm accounted for 98.9% of the HS-IUGR group (Fig. 3b). The results of our study showed that the average particle size of exosomes in IUGR lambs was significantly decreased ($P = 0.001$, Fig. 3c). In addition, our western blot analysis revealed that the extracellular vesicles expressed CD81 protein, which is a surface marker protein of exosomes. Calnexin is an endoplasmic reticulum-related protein, which accelerates protein folding and assembly. Compared to 293T cells, there was no Calnexin protein in exosomes. This finding indicated that the separated vesicles were indeed exosomes and free of somatic cell contamination derived from umbilical plasma in both IUGR and normal lambs, as demonstrated in Fig. 3d. The Micro BCA protein assay was utilized to compare the protein levels of exosomes that were isolated from umbilical plasma. The results indicated that there was no significant difference in the concentration of exosomal protein between the CON and HS-IUGR groups ($P > 0.05$, Supplementary Table 1).

**Small RNA sequencing and small RNA distribution**. To determine the concentration of exosomal RNA, the Quantus Fluorometer was used, and it was found that the total RNA concentration of exosomes in umbilical plasma-derived exosomes from IUGR and normal lambs ranged between 0.16 and 0.60 ng/μl. Figure 4a displays the analysis process for the small RNA sequencing. The sequencing base quality met Q30, which was suitable for subsequent analysis (Fig. 4b). The length of small RNA ranged from 17 to 147 bp (Fig. 4c). The original reads had a total length of 93,610,573, with an average reading length of 17,833,856.3

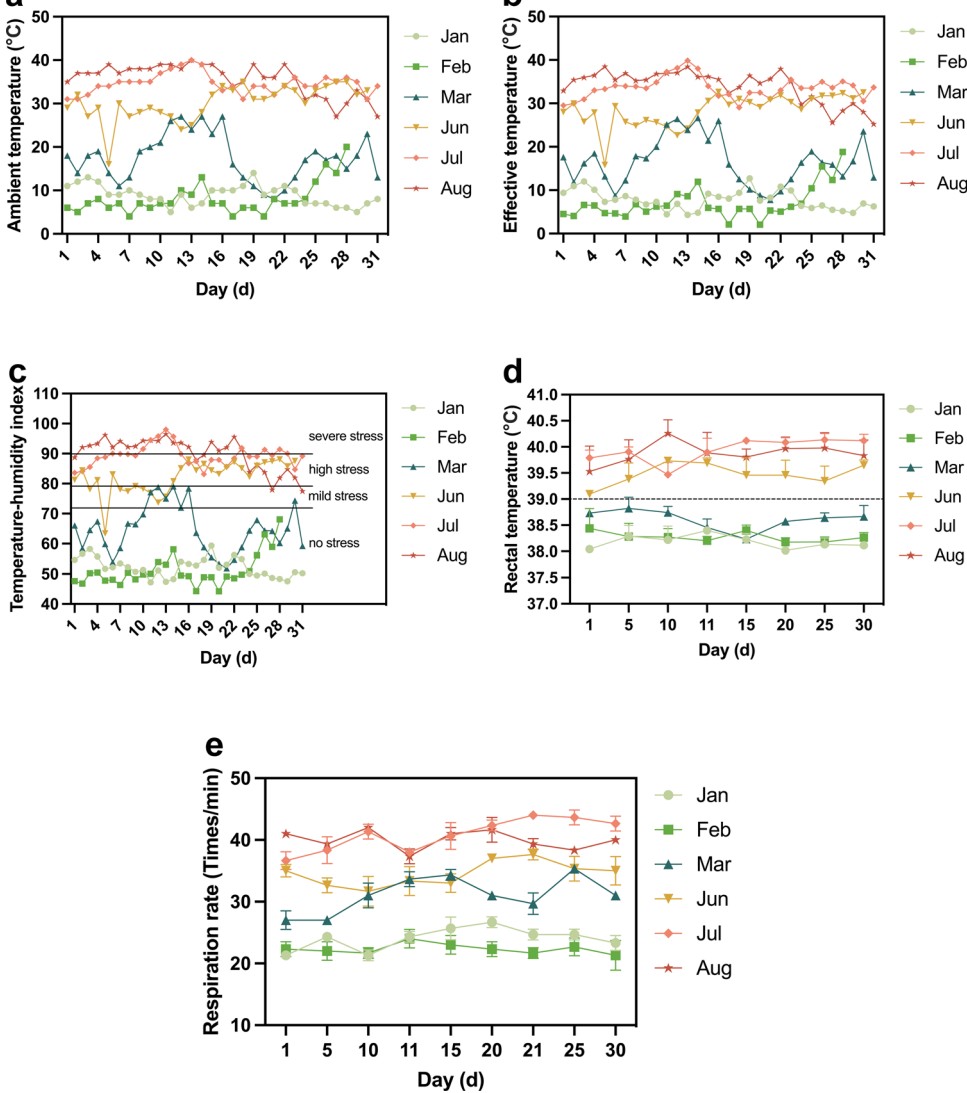

**Fig. 1 Establishment of the heat-stressed model in Hu sheep during pregnancy. a** Ambient temperature of pregnant Hu sheep shed. **b** Effective temperature of pregnant Hu sheep. **c** THI of pregnant Hu sheep. **d** Rectal temperature of pregnant Hu sheep ($n = 10$, ten Hu sheep in each group). **e** Respiration rate of pregnant Hu sheep ($n = 10$, ten Hu sheep in each group). The values were presented as mean ± SEM.

for the CON group and 13,369,668 for the HS-IUGR group (Supplementary Table 2). After quality filtering, a large proportion of raw reads remained, with clean_GC accounting for over 50% of raw_GC (Supplementary Table 2). This suggested that the sequencing quality of small RNA from exosomes derived from umbilical plasma in both the CON and HS-IUGR groups was high. To remove ncRNA from the clean reads, including rRNA, snoRNA, sRNA, and lncRNA, which were compared to the Rfam library and discarded any matches. Then the remaining reads were mapped to the sheep genome, and the mapped rate was obtained (Supplementary Table 3). The comparison of Rfam revealed that rRNA made up the largest portion of ncRNA, as shown in Fig. 4d. Furthermore, the sequencing data indicated that the majority of miRNAs in exosomes from umbilical plasma were 21–22 nt in length, as demonstrated in Fig. 4e.

**Differential expression analysis of exosomal miRNAs.** In total, 71 miRNAs were identified in the exosomes of umbilical plasma from both the CON and HS-IUGR groups. In addition, 62 and 63 miRNAs were expressed in exosomes from the umbilical plasma of normal and IUGR Hu lambs, respectively.

Nevertheless, only two miRNAs showed significant expression when compared to sheep precursors in miRBase, based on the screened standard of |log2 Fold Change| ≥ 1 ($P < 0.05$, Fig. 5a). There were 42 common miRNAs between the CON and HS-IUGR groups, with eight unique miRNAs in CON and nine in HS-IUGR (Fig. 5b). Notably, oar-miR-411a-5p was significantly downregulated and oar-miR-200c was significantly upregulated in HS-IUGR compared to the CON ($P < 0.05$, Fig. 5c). In Fig. 5d, the log2 Fold Change of differentially expressed miRNAs was presented.

**Target genes prediction of differentially expressed miRNAs.** To predict the target genes of oar-miR-200c and oar-miR-411a-5p, which were differentially expressed miRNAs found in exosomes derived from umbilical plasma, MiRanda was used. The analysis revealed 1700 and 1029 unique target genes for oar-miR-200c and oar-miR-411a-5p, respectively. Interestingly, there were 291 common genes between the two groups, as shown in Supplementary Fig. 1. The common genes of oar-miR-200c and oar-miR-411a-5p were illustrated in Supplementary Fig. 1.

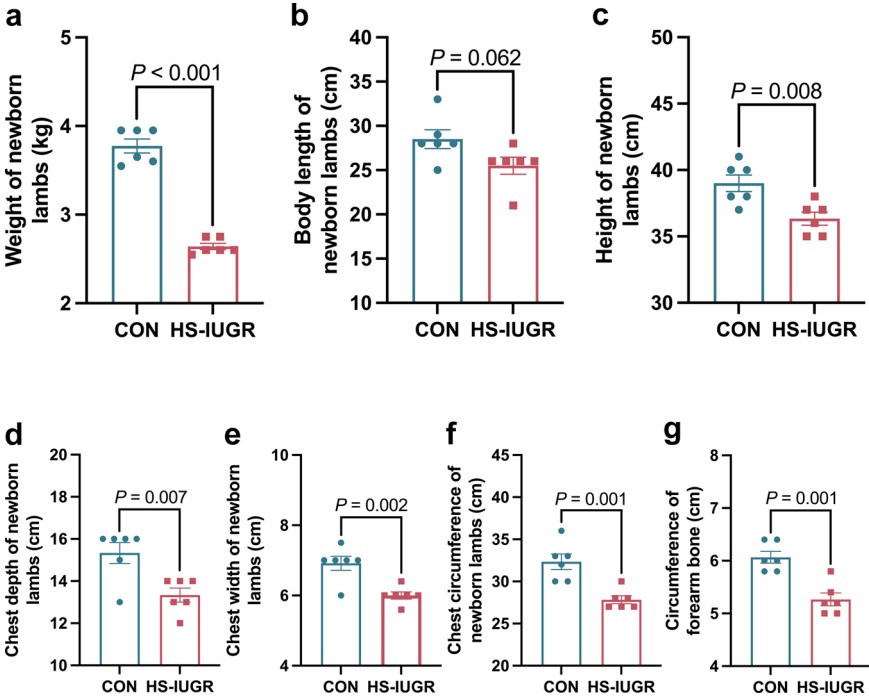

**Fig. 2 Body measurement traits of the normal and IUGR lambs (*n* = 6, six lambs in each group). a** Weight of the normal and IUGR lambs. **b** Body length of the normal and IUGR lambs. **c** Height of the normal and IUGR lambs. **d** Chest depth of the normal and IUGR lambs. **e** Chest width of the normal and IUGR lambs. **f** Chest circumference of the normal and IUGR lambs. **g** Circumference of forearm bone in the normal and IUGR lambs. The values were presented as mean ± SEM.

**GO and KEGG enrichment analysis of differentially expressed miRNAs**. GO and KEGG enrichment analyses were conducted to determine the roles of target genes in various biological processes (Fig. 6). The findings revealed that 291 target genes were commonly enriched in 68 GO terms, including 18 terms related to biological processes, 24 terms related to cellular components, and 26 terms related to molecular functions. Notably, these target genes were significantly enriched in several key pathways, such as the Wnt signaling pathway, cell adhesion, and cytoskeleton organization (*P* < 0.05, Fig. 6a). Furthermore, the target genes exhibited significant enrichment in 35 KEGG pathways, such as the cAMP, TGF-beta, calcium, and Rap1 signaling pathways (*P* < 0.05, Fig. 6b).

**Validation of differentially expressed miRNAs**. To confirm the precision of small RNA-seq and investigate the role of miRNAs in the development of Hu lambs, RT-qPCR was performed to assess the expression levels of differentially expressed miRNAs, specifically oar-miR-411a-5p and oar-miR-200c. The findings of this study demonstrated that the expression of oar-miR-411a-5p was significantly reduced, while oar-miR-200c was significantly increased in HS-IUGR compared to the CON (*P* < 0.05, Fig. 7b). Furthermore, the expression levels of the differentially expressed miRNAs were consistent with the small RNA-seq results, which suggested that the sequencing accuracy was high (Fig. 7a).

**Discussion**
Global warming, an increasingly prevalent climate phenomenon, has a detrimental effect on the health and productivity of both humans and animals due to continuous hyperthermia[20]. To gain a better understanding of this impact, we have conducted various studies on heat stress and exosomes in vivo and in vitro[19,21,22]. Hu sheep are a locally protected breed of livestock and poultry in China, originating from the Tai Lake basin in both Jiangsu and Zhejiang provinces. They are known for their long estrus cycle and high litter size[23]. However, heat stress can negatively impact the reproduction of Hu sheep and may lead to the occurrence of IUGR. Therefore, we chose Hu sheep as our experimental animals. To evaluate the levels of heat stress, we used THI, a common index that utilizes climate parameters, which is used to measure the level of heat stress in animals. When THI is less than 72, there is no stress. If THI falls between 72 and 79, there is mild stress. High stress occurs when THI ranges from 80 to 90, and severe stress is indicated when THI exceeds 90[24]. In this study, pregnant Hu sheep were found to be in a no-heat-stressed state for the majority of the time, with only a few days in January, February, and March showing mild stress. However, during June, July, and August, pregnant Hu sheep experienced high and even severe stress. To evaluate the degree of heat stress, rectal temperature, and respiration rate are considered key indicators[25]. The study found that heat-stressed Hu sheep had higher rectal temperatures and respiration rates compared to non-heat-stressed Hu sheep. These results suggest that a model for heat-stressed Hu sheep during mid-late gestation has been established.

To induce IUGR animal models, researchers altered the ambient temperature. For example, Limesand et al. (2006) increased the ambient temperature of ewes during late pregnancy to obtain IUGR lambs[9]. This study found that exposing Hu sheep to heat stress during mid-late gestation resulted in the birth of IUGR lambs. The improper prenatal environment caused cellular stress and death in the placenta, which in turn hindered fetal growth[26]. The dysfunction of the placenta led to IUGR, as there were discrepancies between the metabolic needs of the fetus and the placental supply[27]. The weight, height, chest depth, chest width, chest circumference, and circumference of the forearm bone were significantly decreased in IUGR lambs compared to normal lambs (*P* < 0.01). This reduction in growth is due to the placenta's inability to supply sufficient nutrients and oxygen to the fetus[28]. In addition, IUGR has long-lasting effects on organ

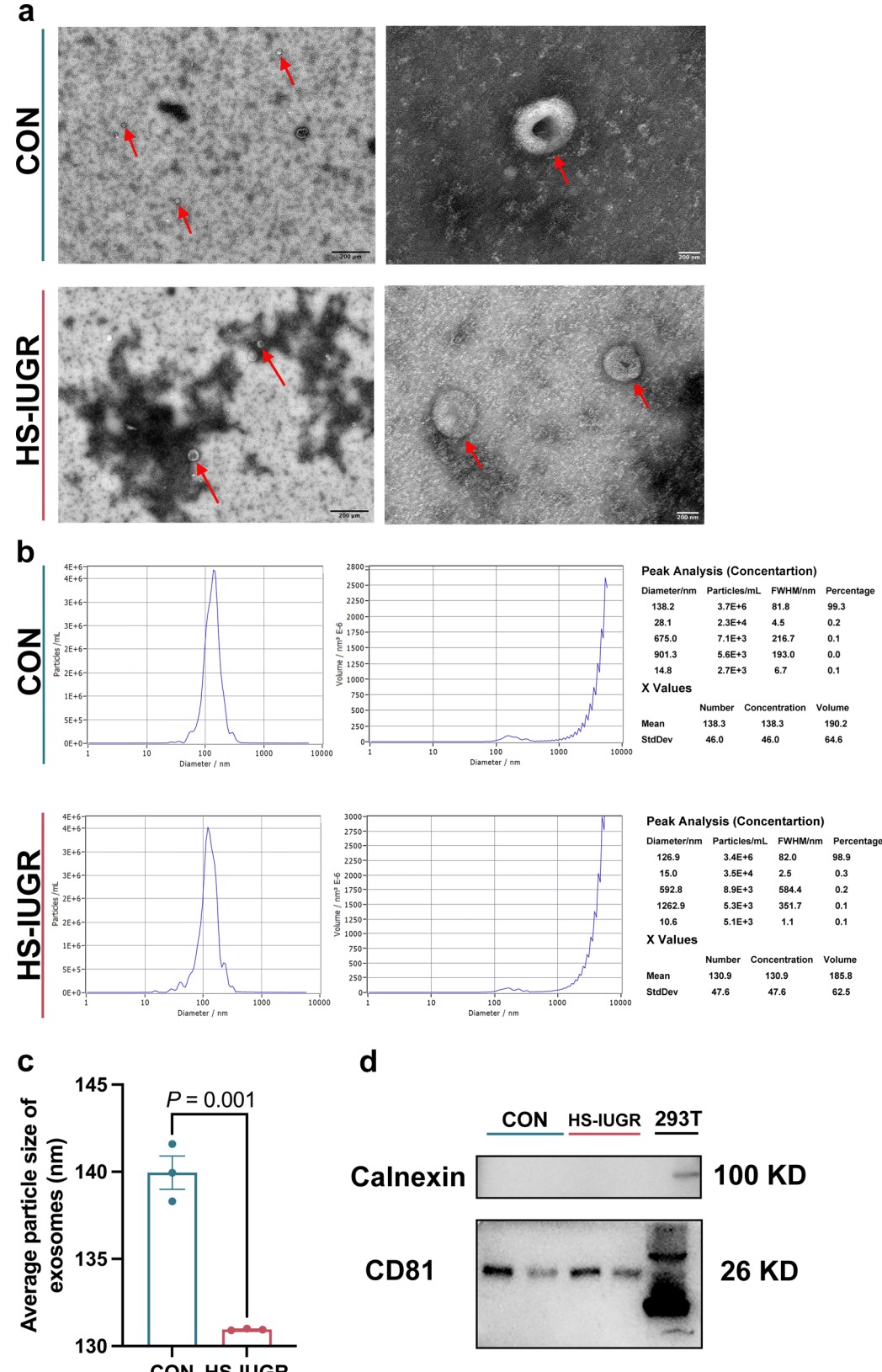

**Fig. 3 Isolation and identification of exosomes from umbilical plasma in normal and IUGR lambs ($n = 3$, three exosome samples in each group).**
**a** Representative images of exosomes separated from umbilical plasma by transmission electron microscopy (TEM) (scale bar, 200 μm, and 200 nm).
**b** Representative photos of exosomes from normal and IUGR lambs by nanoparticle tracking analysis, demonstrating the diameter and particle distribution of separated exosomes. **c** Analysis of exosomal average particle size. **d** Representative images of exosomal markers CD81 by western blot. The values were presented as mean ± SEM.

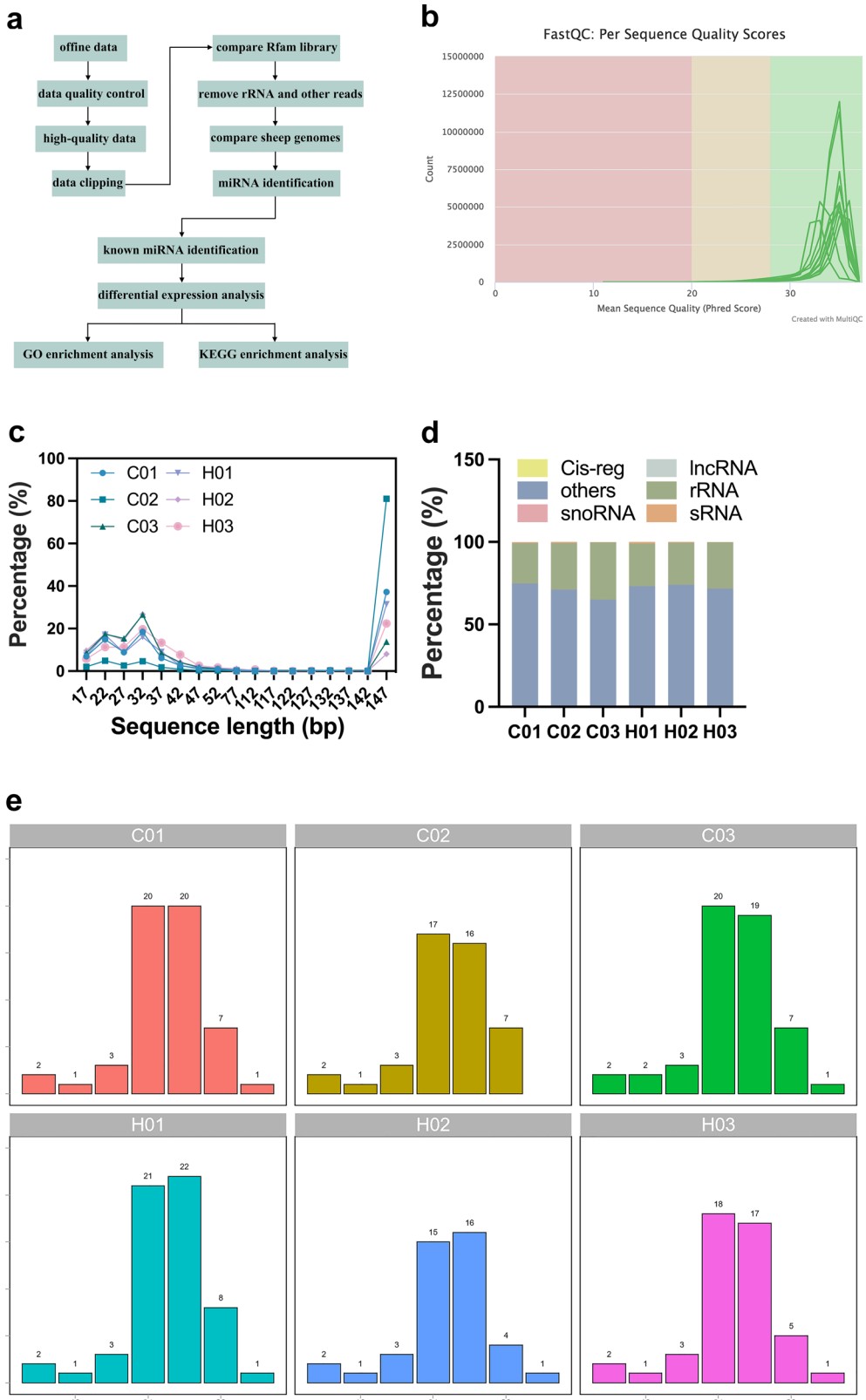

**Fig. 4 Small RNA sequencing results of exosomes derived from umbilical plasma in normal and IUGR lambs ($n = 3$, three exosome samples in each group). a** Analysis process of small RNA sequencing. **b** Quality distribution of small RNA sequencing. **c** Distribution of small RNA length in exosomes. **d** Analysis of the clean reads by Rfam. **e** Distribution of expressed miRNA length.

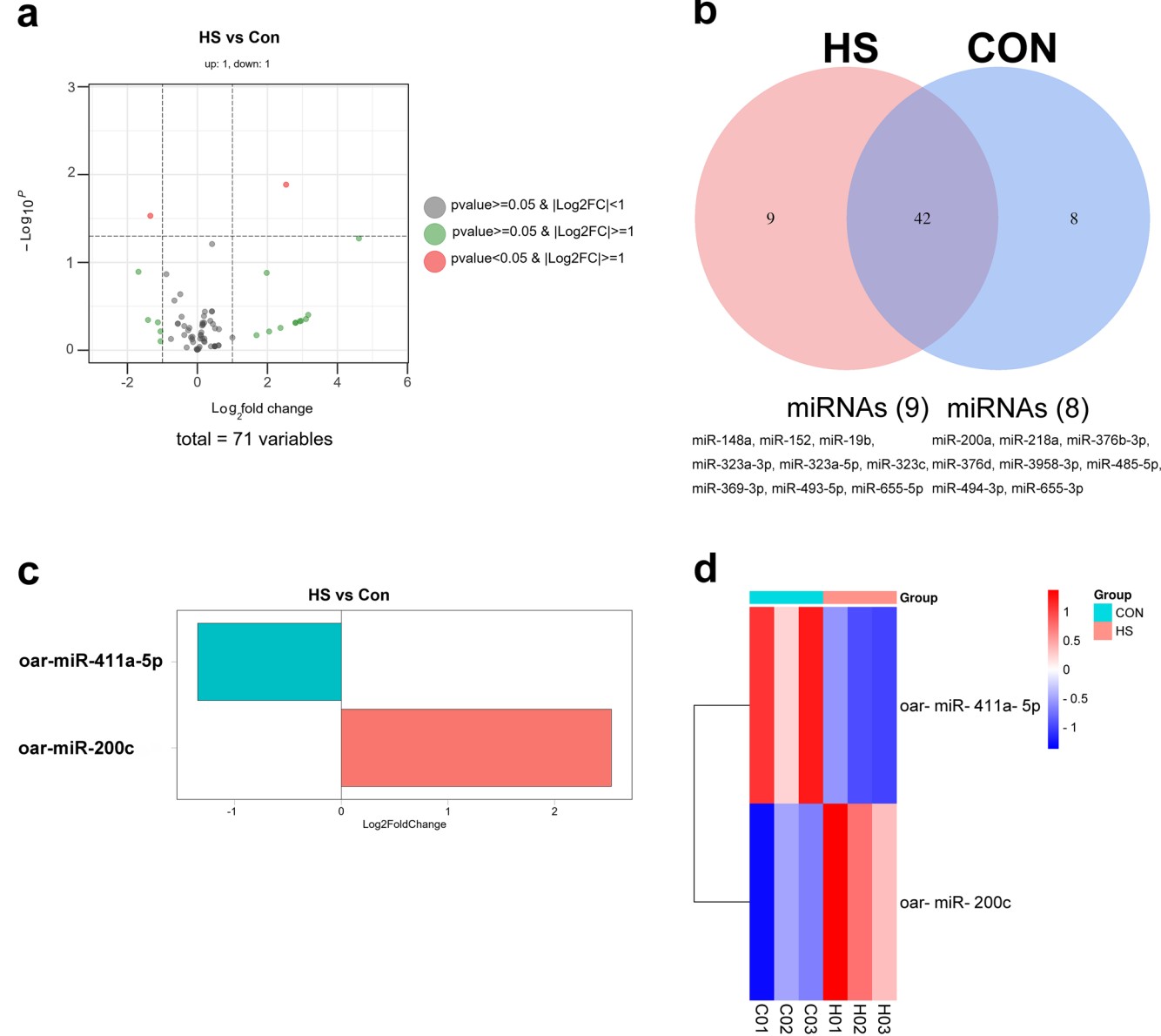

**Fig. 5 Differentially expressed miRNAs of exosomes originated from umbilical plasma in CON and HS-IUGR groups ($n = 3$, three exosome samples in each group). a** Volcanic map of miRNAs. The same miRNA may derive from different predecessors, so some points coincide. **b** Venn diagram shows the common and unique miRNAs of CON and HS-IUGR groups. **c** Heatmap of differentially expressed miRNAs. **d** Histogram of differential miRNA expression. The $Y$ axis shows mature miRNA/precursor miRNA.

function[26]. There were other differences between the normal and IUGR lambs, which might be resulted from genetic factors.

Several studies have utilized sheep as a model for in vitro experiments related to exosomes. These studies have revealed a correlation between exosomes and the development of sheep embryos. Specifically, researchers have observed differences in the expression of mRNA, miRNA, and proteome in the extracellular vesicles of uterine fluid between pregnant sheep at 14 days and non-pregnant sheep. Furthermore, it has been discovered that exosomes secreted by the ovine uterus can be absorbed by both embryos and endometrium[29,30]. In sheep, the acetylcholinesterase content in serum, which represents the vitality of exosomes, remained stable during mid-gestation. However, it decreased in pregnant sheep on gestational day 133[31]. This suggests that the content of exosomes in serum changes during pregnancy. Furthermore, the miRNAs of exosomes from serum, umbilical serum, and placentomes were sequenced, and it was discovered

that miRNAs found in serum exosomes can target cell growth, proliferation, and organ development pathways[19]. In addition, miRNAs found in exosomes from umbilical serum and placentomes were found to target cellular, biological, and embryonic development signaling pathways[19]. These findings demonstrate that exosomes play a crucial role in ovine pregnancy by facilitating direct and dynamic communication between the embryo and the maternal environment. This study found that IUGR lambs had a reduced average particle size and the number of exosomes, suggesting that maternal heat stress inhibited exosomal biogenesis and function in the umbilical cord blood of IUGR lambs, ultimately leading to the occurrence of IUGR.

Exosomes play a crucial role in modulating reproduction, pregnancy, and embryo development in both humans and animals[16]. Exosomes are small vesicles that contain a variety of molecules, including mRNAs, proteins, and non-coding RNAs such as miRNAs, lncRNAs, and circRNAs[32]. These vesicles play a

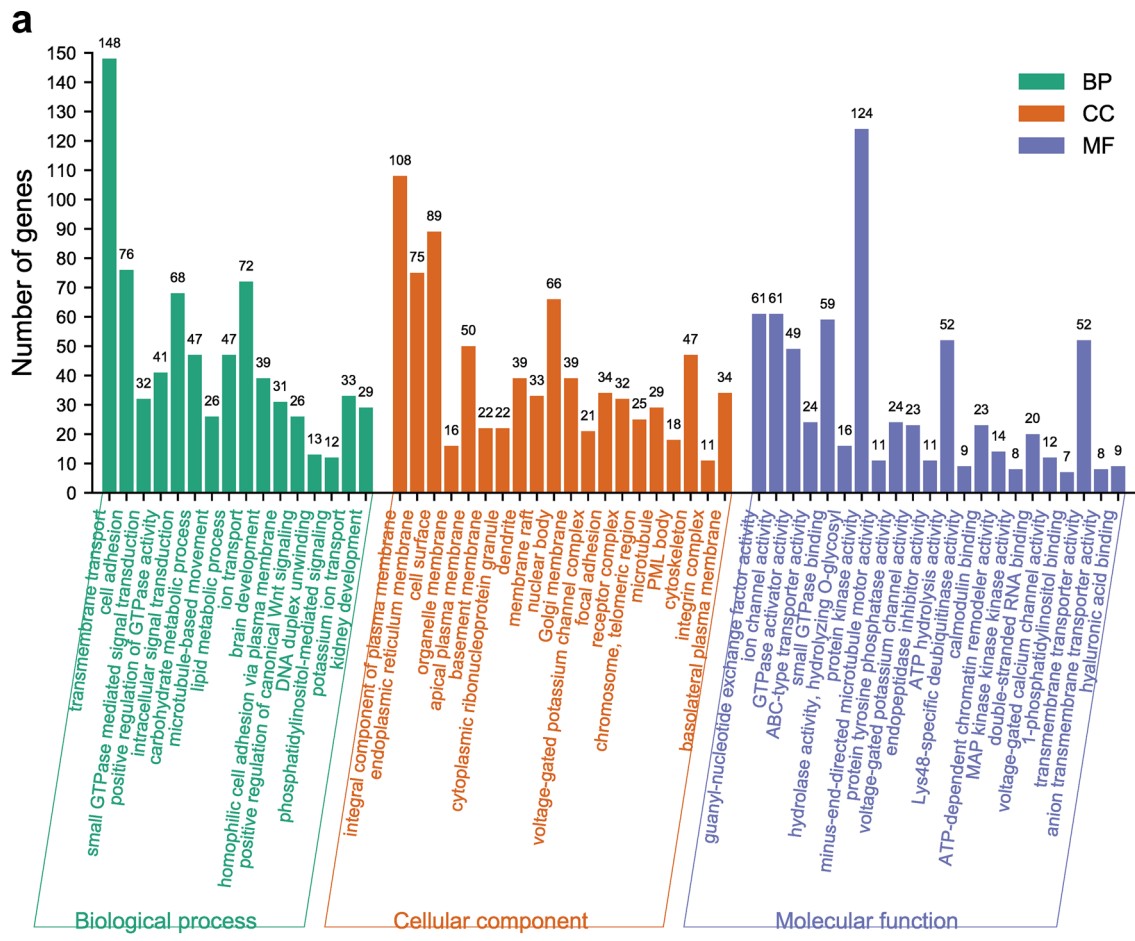

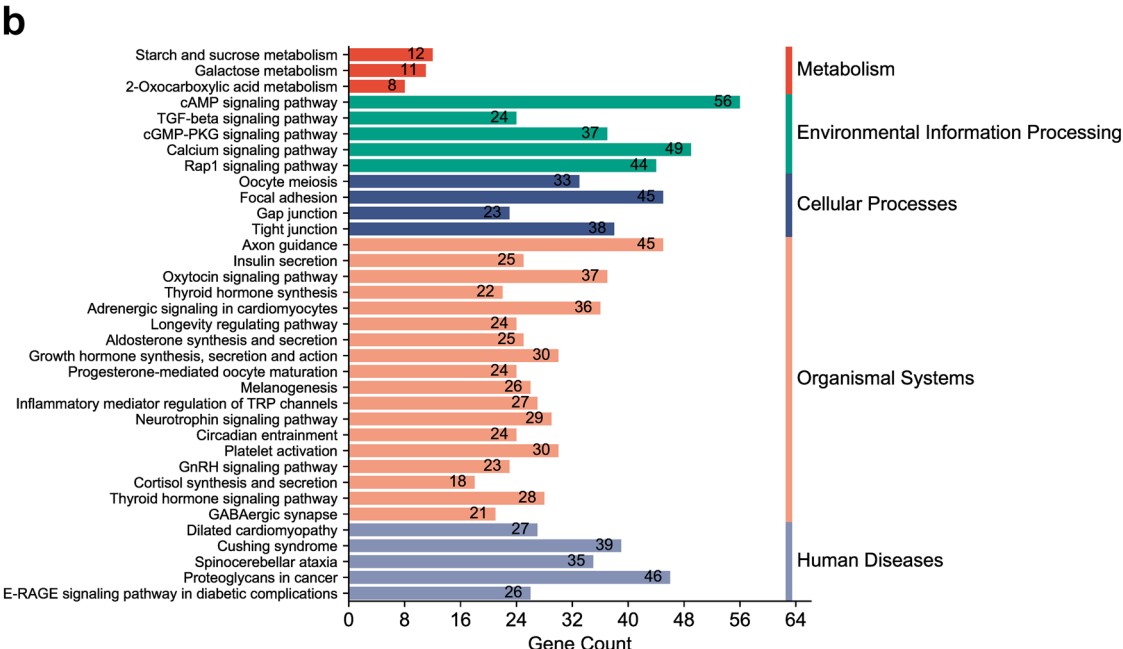

**Fig. 6 Enrichment analysis of common target genes of oar-miR-200c and oar-miR-411a-5p. a** GO enrichment analysis. **b** KEGG enrichment analysis.

crucial role in mediating biological processes by facilitating cell-to-cell communication as conduction molecules[33–36]. Recent studies have shown that miRNAs are distributed in various body fluids, including blood, saliva, urine, sperm, and milk, as revealed by small RNA sequencing[37]. Exosomal miRNAs have been found to regulate the development of mammalian embryos, with miR-766-3p, miR-663b, and miR-132-3p from exosomes having a significant impact on embryo development[38]. Exosomes derived

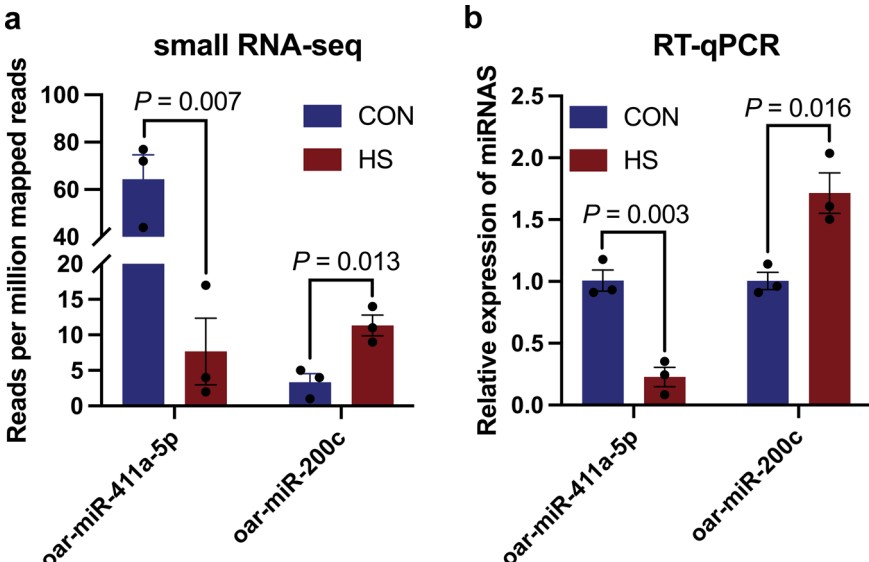

**Fig. 7 Expression analysis of significantly expressed miRNAs in exosomes derived from umbilical plasma of normal and IUGR lambs ($n = 3$, three exosome samples in each group). a** Expression of oar-miR-411a-5p and oar-miR-200c in exosomes detected by small RNA-seq. **b** Relative expression level of oar-miR-411a-5p and oar-miR-200c in exosomes determined by RT-qPCR. The values were presented as mean ± SEM.

from follicular fluid contain miR-134, miR-323-3p, and miR-410, which play a role in embryo development[39,40]. When the scrotum is exposed to heat stress, the miRNA content of small extracellular vesicles decreases, leading to an impact on the function of the testis and epididymis, and miR-126-5p may be transferred between small extracellular vesicles and sperm, which is associated with spermatogenesis and maturation of bovine sperm[41]. Up to this point, no studies have reported on the expression of exosomal miRNAs derived from umbilical plasma between IUGR and normal lambs. We hypothesize that heat stress alters the composition of exosomal miRNAs from umbilical plasma. In this study, we aimed to demonstrate the differential expression of exosomal miRNAs from umbilical plasma between IUGR and normal lambs. We identified 71 miRNAs including two differentially expressed miRNAs of exosomes derived from umbilical plasma in both IUGR and normal lambs. There were 13 significantly expressed miRNAs of exosomes in the umbilical arterial serum compared to the umbilical venous serum of lambs[19]. 116 and 226 miRNAs of umbilical cord blood were differentially expressed in IUGR piglets and IUGR fetuses[42]. The differentially expressed miRNAs were less than the previous studies, the discrepancy may be due to the difference in species. Specifically, oar-miR-411a-5p was significantly downregulated and miR-200c was significantly upregulated in HS-IUGR ($P < 0.05$). This difference in expressed miRNAs is in accordance with the cellular function[43], indicating a potential role in the pathogenesis of HS-IUGR.

The GO enrichment analysis elucidated that the target genes were enriched in a total of 68 GO terms. Of these, 18 were related to biological processes, 24 were related to cellular components, and 26 were related to molecular functions. In addition, the KEGG pathway enrichment analysis showed that the target genes were enriched in several pathways, including the Wnt, cAMP, TGF-beta, calcium, and Rap1 signaling pathways. The signaling pathways identified in previous studies have been shown to affect the development of skeletal muscle[44–48]. Based on this knowledge, we hypothesized that heat stress experienced during pregnancy may have altered the miRNAs present in exosomes found in umbilical plasma. This alteration could then have led to the modulation of target genes, ultimately disrupting the signaling pathways and impairing the development of skeletal muscle in

IUGR lambs. However, this hypothesis was not effectively tested in this study.

The differences in the umbilical cord blood are a consequence of IUGR. During pregnancy, miRNAs derived from umbilical cord blood make a difference in fetal development[42]. In a study conducted by Luo et al.[42], it was found that there were differences in the exosomal miRNAs of umbilical cord blood between IUGR and normal piglets, as confirmed by transcriptome sequencing. These findings are consistent with our results[42]. Therefore, further exploration of exosomes derived from umbilical cord blood is necessary to shed light on the relationship between miRNA expression and fetal development under heat stress.

In conclusion, the heat stress experienced by Hu sheep during mid-late gestation resulted in the occurrence of IUGR. Exosomes derived from umbilical plasma were isolated and identified from both IUGR and normal lambs, and a small RNA sequencing was conducted to detect the expression of exosomal miRNAs. Heat stress has been found to alter the expression of exosomes derived from umbilical plasma. Specifically, two miRNAs, oar-miR-411a-5p and oar-miR-200c, have been identified as potentially having a significant impact on fetal development in heat-stressed Hu sheep through various signaling pathways. This study provides new insights into the potential role of exosomal miRNAs in the pathogenesis of IUGR and the effects of heat stress on fetal development.

## Methods

**Inclusion and ethics.** All experimental procedures were approved by the Animal Care and Use Committee of Nanjing Agricultural University and were strictly conducted according to the animal experiment guidelines (Approval ID: SYXK 2022-0031; Approval Date: 2022-6-10). We have complied with all relevant ethical regulations for animal testing.

**Determination of environmental indicators and thermal parameters.** The temperature in the sheep shed was measured daily during the months of January, February, March, June, July, and August using a dry and humidity thermometer. Both dry bulb temperature (Td) and wet bulb temperature (Tw) were recorded. The effective temperature (ET) and temperature-humidity index

(THI) were calculated using the formula[24]:

$$THI = 0.72 \times (Td + Tw) + 40.6, ET = 0.35Td + 0.65Tw$$

In addition, rectal temperature and respiration rate were measured using the method[49]. To obtain the rectal temperature of the pregnant Hu sheep, an animal thermometer was inserted approximately three to five cm into the rectum. After waiting for 5 min, the temperature value was read. Additionally, the respiratory rate, indicated by the fluctuation frequency of the lumbar fossa, was observed during a fixed time while the Hu sheep were at rest. The rectal temperature and respiration rate were measured on the 1st, 4th, 7th, 10th, 13th, 16th, 19th, 22nd, 25th, 28th, and 31st (30th) in Jan, Feb, Mar, Jun, Jul, and Aug.

**Sample collection**. The compatriots or half-compatriots grown-up and healthy female Hu ewes were selected with the same parity and age stage using synchronous estrus and artificial insemination techniques so that the Hu fetuses were born in March and August of the same year, respectively. The pregnant ewes were fed under the same management conditions at Taicang sheep farm, Jiangsu province, China. The pregnant ewes were fed the silage, and the nutritional composition of the silage is shown in Supplementary Table 4. The pregnant Hu sheep had ad libitum access to feed and water throughout the gestation period. Each pregnant Hu sheep was raised in a single pen. The body condition of the pregnant ewes was observed every day. The weight of the lambs was measured upon completion of the pregnancy, and IUGR lambs were selected based on a standard that required their weight to be lower than two standard deviations of the average weight for their age. The umbilical venous blood samples were collected immediately after the Hu lambs were delivered from two groups: six normal male lambs (CON group) born in March, and six IUGR male lambs (HS-IUGR group) born in August. EDTA2K blood collection tubes were used to collect the blood samples, which were then mixed with an anticoagulant by inverting the tubes five times. After collection, the umbilical venous blood was centrifuged at $1900 \times g$ for 10 min at 4 °C by a small refrigerated centrifuge (Beckman, Microfuge 20 R), followed by a second centrifugation at $3000 \times g$ for 15 min at 4 °C. The separated plasma from each sample was collected and stored in a 1.5-mL centrifugal tube at −80 °C for future use.

**Body measurement traits of the Hu lambs**. Various physical characteristics of both normal and IUGR lambs were measured, including body length, height, chest depth, chest width, chest circumference, and forearm bone circumference.

**Extraction and identification of exosomes**. To extract exosomes from umbilical plasma, an ultracentrifuge (Hitachi, CP100MX) was utilized. The frozen plasma sample (4 mL) was thawed at 37 °C, and the resulting solution was transferred to a new tube and centrifuged at $2000 \times g$ for 30 min at 4 °C. The supernatant was carefully transferred to a new tube and centrifuged at $10,000 \times g$ for 45 min at 4 °C. The resulting supernatant was then filtered using a 0.45-μm filter membrane, and the filtrate was collected. Next, the filtrate was transferred to a new centrifuge tube and centrifuged with an overspeed rotor at $100,000 \times g$ for 70 min at 4 °C. The pellet was resuspended in 10 mL of precooled 1× PBS. The overspeed rotor was utilized for another round of centrifugation at $100,000 \times g$ for 70 min at 4 °C. Exosomes were resuspended with 300 μL of precooled 1× PBS before being stored at −80 °C ultra-low-temperature freezer (Thermo, 905) for downstream applications.

**RNA isolation**. Exosomes (100 μL) were mixed with RNA lysate (700 μL). Following this, chloroform (140 μL) was added and vortexed for 15 s. The tubes were then incubated at room temperature for 3 min before being centrifuged at $12,000 \times g$ for 15 min at 4 °C. The upper aqueous phase was transferred to a new EP tube and mixed with 525 μL of absolute ethyl alcohol in a 1.5-ml tube using a pipette (Eppendorf, Research Plus). Next, a mixed solution containing all precipitates was transferred to the RNeasy adsorption column and centrifuged at $8000 \times g$ for 15 s. The filtrate was then discarded, and this step was repeated for the remaining mixture. Buffer RWT (700 μL) and Buffer RPE (500 μL) were used to wash the adsorption column, which was centrifuged at $8000 \times g$ for 15 s, and the filtrate was discarded, respectively. Buffer RPE (500 μL) was used to wash the adsorption column, centrifugation at $8000 \times g$ for 2 min. Then the adsorption column was transferred to a new 2-mL centrifuge tube, centrifuged at $12,000 \times g$ for 1 min to dry, and the filtrate and collecting tube were discarded. Finally, the adsorption column was transferred to a new 1.5-mL centrifuge tube. Then, RNase-free water (30 μL) was added to the adsorption membrane, and the mixture was centrifuged at $8000 \times g$ for 1 min to elute the RNA. To determine the concentration of RNA, the extracted RNA (1 μL) was stained and analyzed using a nucleic acid analyzer (Promega, Quantus Fluorometer). The RNA was then stored at −80 °C.

**Transmission electron microscopy**. A 10-μL exosome sample was dripped onto a 200-mesh copper mesh (Beijing Zhongjing-keyi Technology Co., Ltd., BZ11022a) and allowed to precipitate for 1 min. Next, 1% uranium acetate (10 μL) was dripped onto the copper mesh and allowed to precipitate for 1 min, with the floating liquid again absorbed by the filter paper. The sample was left to drip for several minutes before being observed under a transmission electron microscope (Hitachi, HT-7700) at 100 kV to obtain images of the exosomes.

**Nanoparticle tracking analysis**. The frozen exosome samples were thawed by immersing them in water at 25 °C and then placed on ice. After diluting the samples with 1× PBS, we used a nanoparticle tracking analyzer (PARTICLE METRIX, ZetaVIEW) to detect them. Each sample was imaged in six photos, including the field of view at 1 μm, 100 nm, 200 nm, and 500 nm. Six samples were analyzed and three duplicates were in each group.

**Cell culture**. The 293 T cell line preserved in our laboratory was cultured with DMEM (Gibco, USA) containing 10% fetal bovine serum (FBS) (Gibco, USA) and 1% penicillin/streptomycin (Gibco, USA). The culture medium was changed every two days. The cells were cultured in a 5% $CO_2$ incubator at 37 °C.

**Protein extraction**. The frozen exosomes were thawed at 37 °C to prepare the exosome samples, and 5× RIPA lysis buffer was added promptly. The mixture was then placed on ice for 30 min. The concentration of the standard sample was determined using the BCA kit, with 5 μL of the sample added to the BCA mixture. After incubation at 37 °C for 30 min, the absorbance value was measured at OD 562 nm using a microplate reader (Thermo, Varioskan LUX). The concentration of the exosome samples was calculated according to the standard curve.

**Western blot**. To create the SDS-PAGE gel, a 12% concentration was used based on the protein size of the sample. The protein samples mixed with the SDS sample loading buffer (5×) (Solarbio, Beijing, China) were boiled at 95 °C in a metal bath for 5 min before being added to the electrophoresis gel along with the

marker (Thermo Pierce, 26616). After electrophoresis, the gel was fixed and a suitable PVDF membrane (absin, Shanghai, China) was cut and activated in methanol for 20 s. The exosome protein was then transferred to the PVDF membrane. After the transfer, the PVDF membrane was blocked with 5% skimmed milk diluted in 1 × TBST (Solarbio, Beijing, China) for one hour. Next, the membrane was cut and incubated overnight at 4 °C in a dilute primary antibody calnexin (Proteintech, 10427-2-AP) with a dilution ratio of 1:5000 and CD81 (SAB, 41779) with a dilution ratio of 1:1000. In all, 1× TBST was used to wash the membrane three times, each time for 10 min. Finally, the membrane was incubated at room temperature for 1 h in a dilute Goat anti-Rabbit IgG Peroxidase Conjugated antibody (Merck Millipore, AP132P) with a dilution ratio of 1:5000. In total, 1× TBST was used to wash the membrane three times, each time for 10 min. Next, an equal volume of mixed ECL A/B solution (Biosharp, Hefei, China) was added to the membrane protected from light for 5 min. Finally, the ultra-sensitive chemiluminescence gel imaging system (CLINX, ChemiScope 3000mini) was used to expose the image.

**cDNA library purification**. To construct the cDNA library, we connected the total RNA of exosomes with a 3' adapter and a 5' adapter and performed reverse transcription polymerase chain reaction (RT-PCR). Next, QMN Beads (143 μL) (Qiagen, 331505) were added to the reverse transcript product, mixed by vortexing for 3 s, and incubated for 5 min after brief centrifugation. The solution was then placed on a magnetic rack, and the supernatant was removed after the magnetic beads had completely adsorbed it. In all, 80% ethanol (200 μL) was added to the mixture. The supernatant was then removed, and this process was repeated once. The remaining liquid was absorbed and dried for 10 min. Next, nuclease-free water (17 μL) was added to cover the magnetic beads. The mixture was then incubated for 2 min after being thoroughly mixed. Once the magnetic beads had completely adsorbed the supernatant, the resulting liquid (15 μL) was transferred to a new centrifuge tube. The cDNA was purified and stored at −20 °C.

**Small RNA sequencing**. To determine the expression of miRNAs in umbilical plasma exosomes of IUGR and normal lambs, small RNA sequencing of the exosomes was conducted using the PE150 sequence. We analyzed six samples of exosomes, which included C01, C02, and C03 of the CON group, and H01, H02, and H03 of the HS-IUGR group. The sequencing library's quality was assessed using fastqc. To improve the quality, fastp was utilized to perform N-base excision, q20 filtration, and adapter excision at both ends of the sequence. The Rfam library was compared to remove ncRNAs, such as rRNA, tRNA, and snoRNA, using the bowtie short sequence alignment tool. Small RNAs were quantitatively analyzed using miRDeep2. The normalized miRNAs were then compared between the CON and HS-IUGR groups. The differentially expressed miRNAs were analyzed by DESeq2 to show the expression level between the two groups.

**Prediction of target genes and GO and KEGG enrichment analysis**. miRNAs bind to target genes by complementary pairing with the 3'UTR sites. To predict the target genes of differentially expressed miRNAs in sheep mRNA, the 3'UTR sequence was analyzed using MiRanda. The selection of predicted target genes was based on a seed length greater than 8 and an energetic score less than $10^{-\Delta\Delta G}$. We analyzed the predicted target genes, focusing on functional enrichment through Gene Ontology (GO)

and Kyoto Encyclopedia of Genes and Genomes (KEGG) enrichment analysis. GO is an internationally standardized system for classifying gene function, which offers a comprehensive vocabulary for describing the attributes of genes and gene products in organisms. This system is dynamic and regularly updated to ensure accuracy and relevance. GO is comprised of three ontologies that describe the molecular function (MF), cell component (CC), and biological process (BP). Each term in GO corresponds to an attribute, and the term is the basic unit of GO. GO function analysis provides annotation on the GO function classification of genes and significant enrichment analysis of genes. To map each term, genes were transferred to the GO database (http://www.geneontology.org/), and the number of genes associated with each term was obtained. A hypergeometric test was conducted to detect enriched GO terms in the genes relative to the sheep genome. The Benjamin–Hochberg method (BH) was utilized to calculate the adjusted P value (Padj), with Padj Δ.05 serving as the threshold for significant enrichment.

To gain a deeper understanding of the biological function of the genes, pathway analysis was conducted. The KEGG database is widely recognized as the main public resource for pathway information and was therefore used as the unit for pathway enrichment analysis. For more information on the KEGG database, please visit http://www.genome.jp/KEGG. A hypergeometric test was utilized to identify pathways that exhibited enrichment in genes when compared to the sheep genome. This pathway enrichment analysis allowed for the identification of the most crucial biochemical metabolism and signal transduction pathways that were involved in the genes. The adjusted P value (Padj) was calculated using the Benjamin–Hochberg method (BH).

**Real-time quantitative PCR (RT-qPCR)**. To ensure the precision of small RNA sequencing, RT-qPCR was conducted to compare the expression levels of exosomal miRNAs that were differentially expressed in the CON and HS-IUGR groups. The miRNA primer sequences were synthesized by Beijing Tsingke Biotech Co., Ltd. (Beijing, China). MiRNA 1st Strand cDNA Synthesis Kit (by stem-loop) (Vazyme, MR101) was used to reverse transcribe the RNA of exosomes and then determined the relative expression level of the differentially expressed miRNAs using RT-qPCR[50]. The expression levels of differentially expressed miRNAs were compared using the $2^{-\Delta\Delta Ct}$ method, with U6 serving as the reference gene. The primer sequences used in the analysis can be found in Supplementary Table 5.

**Statistics and reproducibility**. The data were presented as Mean ± SEM and analyzed using SPSS 23.0 (SPSS Inc., Chicago, IL, USA). One-way analysis of variance (ANOVA) was conducted to compare the expression between the CON and HS-IUGR groups. The figures were generated using GraphPad Prism 9 (La Jolla, CA, USA). A level of $P < 0.05$ was considered statistically significant.

**Reporting summary**. Further information on research design is available in the Nature Portfolio Reporting Summary linked to this article.

## Data availability

The raw sequence data reported in this paper have been deposited in the Genome Sequence Archive[51] in National Genomics Data Center[52], China National Center for Bioinformation/Beijing Institute of Genomics, Chinese Academy of Sciences (GSA: CRA012313) that are publicly accessible at https://ngdc.cncb.ac.cn/gsa. The source data behind the graphs can be found in Supplementary Data 1.

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

## Acknowledgements

The authors acknowledge Suzhou Panomix for providing help. https://www.panomix.com/. This study was supported by the National Natural Science Foundation of China (No. 32172725), Ningxia Natural Science Foundation of China (2023AAC02013), and Jiangsu Province Innovation and Entrepreneurship Program (JSSCRC2021569).

## Author contributions

J.L. wrote the manuscript. H.L. planned the experiments. X.Z. provided the funding. Y.L. edited the manuscript. P.Z. carried out the experiments.

## Competing interests

The authors declare no competing interests.
