## [Peer Review File · Communications Biology]

Reviewers' comments:

Reviewer #1 (Remarks to the Author):

The manuscript by Lu et al provides some interesting insights into the regulatory mechanisms regarding the development of heat stress induced IUGR in sheep. However, there are many details missing from the manuscript and this reviewer has some concerns.

-It is well-established that there are major differences between lambs born at different times of year. How can you be sure that these differences are only to do with heat stress?

-You have specifically selected IUGR heat stressed lambs in this study. What is the incidence of IUGR in your animals that lamb in August? Why were no non-IUGR lambs from August added as an additional control?

-Many instances throughout the manuscript where references are needed for the statements in text e.g.

Line 47-49, 52-53, 62-63, 66-67, 453-454,

-Multiple references throughout the paper to the 'parent'. Parent could be mother or father. In most cases I am assuming the authors are referring to the ewe but please take care to correct this throughout.

-Line 42 change 'sheep' to 'ewes'. Expand upon the 'change in fetal morphology'

-Line 45 – 'during the late pregnancy period' – in line 40 you state that it starts to have an effect in mid-gestation. Contradictory statements.

-Line 50-61 – Difficult to follow when reading as changing between sheep and pigs. Also, pigs are usually considered a naturally occurring example of IUGR so need to state if these are experimentally induced models in pigs or naturally occurring. The references to the muscle development phenotype in this section (while correct and interesting) may not be hugely relevant as you haven't looked at anything muscle related in these animals in this study.

-Line 71- change 'which are derived from the placenta' to 'can also be produced by'.

-Line 71 – 'exosomes serve as a marker for pregnancy diagnosis'. They have the potential to be utilized in this way but we do not know enough about the temporal changes in miRNAs in the exosomes in blood yet. Please change terminology as this is a bold statement.

-Line 74 – 'concentration of serum' I assume this is in maternal human serum? But please specify.

-How often was the rectal temperature and respiratory rate quantified?

-Within cohort (march or august lambs) how similar were the breeding/lambing dates?

-How soon after birth was the cord blood sample taken? Were they from the umbilical vein or artery?

-Line 113 – what temperature was the centrifugation at?

-Provide information of the ultracentrifuge used?

-Line 138 – Eppendorf is a brand....not an item. Was this done with a pipette in a 1.5ml tube or something similar? Please specify.

-Line 142 – I believe it is RW1 buffer not RWT

-Line 153 – more information needed regarding the copper mesh

-Line 154 – concentration of the uranium acetate needed

-Line 161 – need a lot more information regarding the nanoparticle tracking analysis

-Line 164 – I think this should be RIPA lysis buffer. RIPA lysate would suggest that this is the lysate in RIPA buffer after protein extraction is complete.

-Western blots – I imagine a loading dye was used – please provide information.

Were the samples boiled in b-mercaptoethanol? What temperature?

What 'marker' was used? Line 175 – change from 'sealed' to 'blocked'.

What was the milk diluted in?

I imagine there were some washes between antibody incubations – please add this information.

Change 'to avoid light' to 'protected from light'

Was a loading control used? Was the membrane stained for total protein?

-Provide information for QMN beads

-Line 191-192 – I assume NF-water is nuclease free water? Please clarify in text

-Line 246 – Did UB6 have stable expression between the groups?

- Line 251 – Change from 'images' to 'Figures were generated'.
- For all figures where appropriate please state what the values presented are e.g. mean +/- SEM
- The n in all figure legends needs clarified. E.g. in figure 1 you have n=10 – is this only 10 ewes total? Or is this 10 ewes that lambed in march and 10 that lambed in august?
- The exosome sizes that you present were approx. 140nm whereas in the introduction you stated that exosomes are between 40 – 100nm. Please address this discrepancy and discuss further.
- How does the 71miRNAs identified in the umbilical plasma exosomes compare with previous literature in ruminants and humans?
- What was the litter size of the animals used in this study? Were all animals used singles? Were any of the animals compared littermates as this would significantly decrease the n of your study and the significance?
- What was the sex of the lambs used?
- What were the ewes fed?
- Line 428-430 – This is a bold statement. If you are stating this you need to refer to papers that show the changes in P4 you are referring to here.
- Line 478 – I do not agree with this sentence. As the samples were taken at term you do not know if it is 'dysfunctional cord blood' that caused the IUGR or if differences in the cord blood are a consequence of IUGR (which is the more likely option)

Reviewer #2 (Remarks to the Author):

The authors highlighted that sheep are vulnerable to heat stress when the temperature rises high degrees. Thus, during the summer the ewes that experience high temperatures lead to heat stress. Additionally, when the sheep is pregnant and exposed to heat stress, it can negatively impact the fetal growth and the weight of the lambs. This condition is known as intrauterine growth restriction (IUGR). They said that IUGR occurs when the weight of the fetus is less than two standard deviations of the average weight or below the 10% of the normal weight of health animals with the same age. Furthermore, the author raised that small vesicles (exosomes) play a key role during cellular communication. It has already been demonstrated that exosomes can represent physiological or pathological conditions. Thus, exosomes could play a critical role in pregnancy diagnosis, embryo implantation and also during the IUGR. In this sense, the authors collected umbilical cord blood from ewes pregnant during the spring (control group - with healthy fetuses) and from ewes during the summer and with IUGR fetus. They isolated the exosomes from the plasma and performed RNA sequencing (small RNA). They found that oar-miR-411a-5p was significantly downregulated in exosomes derived from umbilical plasma of IUGR lambs, while oar-miR-200c was significantly upregulated in the HS-IUGR group. When they performed GO and KEGG enrichment analysis observed that the target genes were involved in the Wnt, TGF-beta, and Rap1 signaling pathway. In general, the results are interesting, the article is well written, however the authors should to answer some points highlighted below.

The IUGR is a condition where animals are born smaller than expected and typically have lower viability and survival rates. Several factors can lead to this condition, as well as, i) under nutrition, heat stress (stated by the authors) and ii) the number and the sex of fetuses per litter. As reported by the authors, Hu sheep are known for their long estrus cycle and high litter size. Did the authors take these other parameters into account? Please, clarify it.

- i) Did the animals are under the same nutritional condition during the spring and summer? If not, how the author guarantee that the IUGR was related to the heat stress and not with nutritional status? How it can interfere in the results?
- ii) Did the number and the sex of the fetuses are the same for both groups (control vs IUGR)?

In the introduction, the authors support very well that pregnant ewes submitted to heat stress can

present IUGR. However, we know that it is not just heat stress that causes this condition. Therefore, authors should add a short paragraph in the introduction mentioning that other factors can also lead to the IUGR condition. In order to keep the introduction not too long, the authors should formulate some paragraphs about heat stress to fit it.

On page 3, line 63 exclude in cells "...produced in cells and secreted by living cells." change it to "produced and secreted by living cells."

The topic Nanoparticle Tracking Analysis in the M&M section should be more detailed. For example, which equipment was used to the analysis? How many records (videos) were made per sample? How many samples were analyzed and were the in duplicates? Etc...

Please, indicate in the text the total volume of plasma utilized by the authors to isolate the exosomes.

On page 12, line 261 there is a mistake in the months of the temperature-humidity index from the summer period. It should be "Jun, Jul and Aug" instead "Jan, Jul and Aug". Please correct it.

I do not know if it is only in my file but the quality of the images from the TEM, the NTA graphs and tables in the Figure 3 are not good and should be improved.

On page 16, Figure 3D the authors used only one exosomal biomarker (CD81). How the authors guarantee there is no contamination from cellular components in the plasma samples? Why the authors did not used a negative marker for exosomes, as well as: GRP78, Calnexin, cytochrome C or Tomm20? One western blot showing no contamination with cellular components (negative marker for exosomes) should be provided by the authors.

In the Western Blot what was the protein concentration used in the 293T cell lysate? This is an excess of protein, which makes it difficult to visualize the band correctly.

In the Venny diagram and Heatmap the names of the miRNAs are too small and unreadable.

On the page 28, lines 475-477 "This alteration could then have led to the modulation of target genes, ultimately disrupting the signaling pathways and impairing the development of skeletal muscle in IUGR lambs." the authors assume that the miRNAs are altering the genes, however they did not test this hypothesis. In this way, add to the end of the sentence. "However, this hypothesis was not effectively tested in this study."

Response to Reviewers Comments

Dear Reviewers,

Thanks very much for taking your time to review this manuscript. We really appreciate all your generous comments and suggestions! According to your advice, all of your questions were answered one by one. Please find my itemized responses below and my revisions in the re-submitted files.

Response to Reviewer 1 comments:

The manuscript by Lu et al provides some interesting insights into the regulatory mechanisms regarding the development of heat stress induced IUGR in sheep. However, there are many details missing from the manuscript and this reviewer has some concerns.

Point 1: It is well-established that there are major differences between lambs born at different times of year. How can you be sure that these differences are only to do with heat stress?

Response 1: Thank you for your reminder, the compatriots or half-compatriots grown-up and healthy female Hu ewes were selected with the same parity and age stage using synchronous estrus and artificial insemination techniques in this study, so that the Hu fetuses were born in March and August of the same year, respectively. The pregnant ewes were fed under the same management conditions at Taicang sheep farm, Jiangsu province, China. Thus, we have minimized the impact of other factors on the fetal sheep as much as possible to make sure that heat stress induced the differences in the birth of Hu lambs.

Line 102-106:

The compatriots or half-compatriots grown-up and healthy female Hu ewes were selected with the same parity and age stage using synchronous estrus and artificial insemination techniques, so that the Hu fetuses were born in March and August of the same year, respectively. The pregnant ewes were fed under the same management conditions at Taicang sheep farm, Jiangsu province, China.

Point 2: You have specifically selected IUGR heat stressed lambs in this study. What is the incidence of IUGR in your animals that lamb in August? Why were no non-IUGR lambs from August added as an additional control?

Response 2: The incidence of IUGR Hu lambs is approximately 30% in August on this sheep farm. It's better to select non-IUGR lambs from August as an additional control, we merely want to explore the differences of miRNAs in exosomes derived from umbilical cord blood in the normal and heat stress-induced IUGR Hu lambs.

Point 3: Many instances throughout the manuscript where references are needed for the statements in text e.g.

Line 47-49, 52-53, 62-63, 66-67, 453-454,

Response 3: Thank you for your suggestion, we have added the references in many instances throughout the manuscript.

Line 50-53:

IUGR is a condition where the weight of the fetus is less than two standard deviations of the average weight for the same age, or below the 10th percentile of the normal weight for the same age, and IUGR is one of the significant complications that can arise during pregnancy [12].

Line 58-59:

Exosomes, which are membranous vesicles with a diameter of 30-150 nm, are produced in cells and secreted by living cells [14].

Line 60-63:

Exosomes are small vesicles that play a crucial role in intercellular communication, and there are around 10^{14} exosomes in the human body, with each cell producing an average of 1000 to 10000 exosomes [16].

Line 470-474:

When the scrotum is exposed to heat stress, the miRNA content of small extracellular vesicles decreases, leading to an impact on the function of the testis and epididymis, and miR-126-5p may be transferred between small extracellular vesicles and sperm, which is associated with spermatogenesis and maturation of bovine sperm [44].

References:

- [12] American College of Obstetricians and Gynecologists' Committee on Practice Bulletins-Obstetrics and the Society for Maternal-Fetal Medicine. ACOG Practice Bulletin No. 204: Fetal Growth Restriction. *Obstet Gynecol.* 2019, 133(2): e97-e109. <https://doi.org/10.1097/AOG.0000000000003070>.
- [14] Théry C, Zitvogel L, Amigorena S. Exosomes: composition, biogenesis and function. *Nat Rev Immunol.* 2002, 2(8): 569-79. <https://doi.org/10.1038/nri855>.
- [16] van Niel G, D'Angelo G, Raposo G. Shedding light on the cell biology of extracellular vesicles. *Nat Rev Mol Cell Biol.* 2018, 19(4): 213-228. <https://doi.org/10.1038/nrm.2017.125>.
- [44] Alves MBR, Arruda RP, Batissaco L, Garcia-Oliveros LN, Gonzaga VH, Nogueira VJM, et al. Changes in miRNA levels of sperm and small extracellular vesicles of seminal plasma are associated with transient scrotal heat stress in bulls. *Theriogenology.* 2021, 161: 26-40. <https://doi.org/10.1016/j.theriogenology.2020.11.015>.

Point 4: Multiple references throughout the paper to the 'parent'. Parent could be mother or father. In most cases I am assuming the authors are referring to the ewe but please take care to correct this throughout.

Response 4: Thank you for your suggestion, we have changed 'parent' into 'ewe'.

Line 54-57:

The impact of ambient temperature on pregnancy varies depending on the stage of gestation, with late pregnancy being more vulnerable than early pregnancy, and there is an almost linear correlation between the ewe's thermal peak and the birth weight of their offspring [11].

Point 5: Line 42 change 'sheep' to 'ewes'. Expand upon the 'change in fetal morphology'

Response 5: Thank you for your suggestion, we have changed ‘sheep’ to ‘ewes’ and described minutely the ‘change in fetal morphology’.

Line 43-46:

During late gestation and in newborn lambs from heat-stressed ewes, there is a change in fetal morphology that the average fetal weight and liver weight decrease, which suggests the presence of intrauterine growth restriction (IUGR) [9].

Point 6: Line 45 – ‘during the late pregnancy period’ – in line 40 you state that it starts to have an effect in mid-gestation. Contradictory statements.

Response 6: Thank you for your reminder, we have changed the statement.

Line 41-43:

Heat stress begins to affect the fetus during mid-gestation, and by late gestation, the size of the fetus is less than two standard deviations from that of a normal sheep [8].

Line 47-49:

This IUGR is caused by the elevated ambient temperature experienced by the ewes during the pregnancy period, and there is a strong correlation between the maximum temperature of the uterus during pregnancy and the birth weight of the offspring [8, 9].

Point 7: Line 50-61 – Difficult to follow when reading as changing between sheep and pigs. Also, pigs are usually considered a naturally occurring example of IUGR so need to state if these are experimentally induced models in pigs or naturally occurring. The references to the muscle development phenotype in this section (while correct and interesting) may not be hugely relevant as you haven’t looked at anything muscle related in these animals in this study.

Response 7: Thank you for your reminder, we have deleted the description of pigs and the muscle development phenotype in this section to make it easier for the readers to follow.

Line 53-57:

Lambs born under severe natural conditions, such as cold and heat stress, experience a decrease in birth weight [13]. The impact of ambient temperature on pregnancy varies depending on the stage of gestation, with late pregnancy being more vulnerable than early pregnancy, and there is an almost linear correlation between the ewe's thermal peak and the birth weight of their offspring [11].

Point 8: Line 71- change ‘which are derived from the placenta’ to ‘can also be produced by’.

Response 8: Thank you for your suggestion, we have changed it.

Line 67-68:

Exosomes can also be produced by the placenta, and have the potential to serve as biomarkers for pregnancy diagnosis by carrying miRNA into the maternal circulation [18].

Point 9: Line 71 – ‘exosomes serve as a marker for pregnancy diagnosis’. They have the potential to be utilized in this way but we do not know enough about the temporal changes in miRNAs in the exosomes in blood yet. Please change terminology as this is a bold statement.

Response 9: Thank you for your reminder, we have changed this sentence.

Line 67-68:

Exosomes can also be produced by the placenta, and have the potential to serve as biomarkers for pregnancy diagnosis by carrying miRNA into the maternal circulation [18].

Point 10: Line 74 – ‘concentration of serum’ I assume this is in maternal human serum? But please specify.

Response 10: Thank you for your suggestion, we have specified it.

Line 70-71:

As gestation progresses, the concentration of serum exosomes from the gestational day of 90 pregnant sheep also increases [20].

Point 11: How often was the rectal temperature and respiratory rate quantified?

Response 11: Thank you for your reminder, we have added this part.

Line 98-100:

The rectal temperature and respiration rate were measured on the 1st, 4th, 7th, 10th, 13th, 16th, 19th, 22nd, 25th, 28th, and 31st (30th) in Jan, Feb, Mar, Jun, Jul, and Aug.

Point 12: Within cohort (march or august lambs) how similar were the breeding/lambing dates?

Response 12: The Hu lambs were born in the second half of March and August. The breeding date of March or August lambs were 148-152 days, and 145-148 days, respectively.

Point 13: How soon after birth was the cord blood sample taken? Were they from the umbilical vein or artery?

Response 13: The cord blood samples were collected immediately after the Hu lambs were delivered. The umbilical artery mainly delivers metabolic waste and carbon dioxide excreted by the fetus to the placenta for treatment by the female parent. The umbilical vein blood of the fetus flows in the umbilical artery. The umbilical vein mainly supplies oxygen and nutrients from the female parent to the fetus. The umbilical artery blood of the fetus flows in the umbilical vein. We focus on the differences of the miRNAs in exosomes from the umbilical venous blood of the IUGR lambs caused by heat stress during the pregnancy of the ewes. Therefore, the umbilical venous blood was collected.

Line 109-112:

The umbilical venous blood samples were collected immediately after the Hu lambs were delivered from two groups: six normal lambs (CON group) born in March, and six IUGR lambs (HS-IUGR group) born in August.

Point 14: Line 113 – what temperature was the centrifugation at?

Response 14: Thank you for your reminder, we have added the temperature for the centrifugation.

Line 113-116:

After collection, the umbilical vein blood was centrifuged at 1900 g for 10 minutes at 4 °C by a small refrigerated centrifuge (Beckman, Microfuge 20R), followed by a second centrifugation at 3000 g for 15 minutes at 4 °C.

Point 15: Provide information of the ultracentrifuge used?

Response 15: Thank you for your reminder, we have provided information of the ultracentrifuge used in this study.

Line 123-124:

To extract exosomes from umbilical plasma, an ultracentrifuge (Hitachi, CP100MX) was utilized.

Point 16: Line 138 – Eppendorf is a brand....not an item. Was this done with a pipette in a 1.5ml tube or something similar? Please specify.

Response 16: We are sorry for this mistake and have corrected it.

Line 138-139:

The upper aqueous phase was transferred to a new EP tube and mixed with 525 µL of absolute ethyl alcohol in a 1.5 ml tube using a pipette (Eppendorf, Research Plus).

Point 17: Line 142 – I believe it is RW1 buffer not RWT

Response 17: It is an RWT buffer.

Point 18: Line 153 – more information needed regarding the copper mesh

Response 18: Thank you for your reminder, we have added the information regarding the copper mesh.

Line 155-157:

A 10 µL exosome sample was dripped onto a 200-mesh copper mesh (Beijing Zhongjingkeyi Technology Co., Ltd., BZ11022a) and allowed to precipitate for one minute.

Point 19: Line 154 – concentration of the uranium acetate needed

Response 19: The concentration of the uranium acetate was 1%.

Line 157-159:

Next, 1% uranium acetate (10 μ L) was dripped onto the copper mesh and allowed to precipitate for one minute, with the floating liquid again absorbed by the filter paper.

Point 20: Line 161 – need a lot more information regarding the nanoparticle tracking analysis

Response 20: Thank you for your reminder, we have added the information regarding the nanoparticle tracking analysis.

Line 164-167:

After diluting the samples with 1 \times PBS, we used a nanoparticle tracking analyzer (PARTICLE METRIX, ZetaVIEW) to detect them. Each sample was imaged in six photos, including the field of view at 1 μ m, 100 nm, 200 nm, and 500 nm. Six samples were analyzed and three duplicates were in each group.

Point 21: Line 164 – I think this should be RIPA lysis buffer. RIPA lysate would suggest that this is the lysate in RIPA buffer after protein extraction is complete.

Response 21: Thank you for your reminder, we have corrected it.

Line 169-170:

The frozen exosomes were thawed at 37 $^{\circ}$ C to prepare the exosome samples, and 5 \times RIPA lysis buffer was added promptly.

Point 22: Western blots – I imagine a loading dye was used – please provide information.

Response 22: Thank you for your reminder, we have provided the information on the loading dye.

Line 178-180:

The protein samples mixed with the SDS sample loading buffer (5 \times) (Solarbio, Beijing, China) were boiled at 95 $^{\circ}$ C in a metal bath for five minutes before being added to the electrophoresis gel along with the marker (Thermo Pierce, 26616).

Point 23: Were the samples boiled in b-mercaptoethanol? What temperature?

Response 23:

Line 178-180:

The protein samples mixed with the SDS sample loading buffer (5×) (Solarbio, Beijing, China) were boiled at 95 °C in a metal bath for five minutes before being added to the electrophoresis gel along with the marker (Thermo Pierce, 26616).

Point 24: What ‘marker’ was used? Line 175 – change from ‘sealed’ to ‘blocked’.

Response 24: Thank you for your suggestion, we have added the information on the ‘marker’ and changed from ‘sealed’ to ‘blocked’.

Line 178-180:

The protein samples mixed with the SDS sample loading buffer (5×) (Solarbio, Beijing, China) were boiled at 95 °C in a metal bath for five minutes before being added to the electrophoresis gel along with the marker (Thermo Pierce, 26616).

Line 183-185:

After the transfer, the PVDF membrane was blocked with 5% skimmed milk diluted in 1 × TBST (Solarbio, Beijing, China) for one hour.

Point 25: What was the milk diluted in?

Response 25:

Line 183-185:

After the transfer, the PVDF membrane was blocked with 5% skimmed milk diluted in 1 × TBST (Solarbio, Beijing, China) for one hour.

Point 26: I imagine there were some washes between antibody incubations – please add this information.

Response 26: Thank you for your reminder, we have added this information.

Line 185-192:

Next, the membrane was cut and incubated overnight at 4 °C in a dilute primary antibody CD81 (SAB, 41779) with a dilution ratio of 1: 1000. 1 × TBST was used to wash the membrane three times, each time for ten minutes. Finally, the membrane was incubated at room temperature for one hour in a dilute Goat anti-Rabbit IgG Peroxidase Conjugated antibody (Merck Millipore, AP132P) with a dilution ratio of 1: 5000. 1 × TBST was used to wash the membrane three times, each time for ten minutes. Next, an equal volume of mixed ECL A/B solution (Biosharp, Hefei, China) was added to the membrane to avoid light for five minutes.

Point 27: Change ‘to avoid light’ to ‘protected from light’

Response 27: Thank you for your suggestion, we have changed ‘to avoid light’ to ‘protected from light’.

Line 191-192:

Next, an equal volume of mixed ECL A/B solution (Biosharp, Hefei, China) was added to the membrane protected from light for five minutes.

Point 28: Was a loading control used? Was the membrane stained for total protein?

Response 28: We used 293T cells as a loading control, and the membrane was stained for total protein.

Point 29: Provide information for QMN beads

Response 29: Thank you for your reminder, we have provided information for QMN beads.

Line 198-200:

Next, QMN Beads (143 μ L) (Qiagen, 331505) were added to the reverse transcript product, mixed by vortexing for three seconds, and incubated for five minutes after brief centrifugation.

Point 30: Line 191-192 – I assume NF-water is nuclease free water? Please clarify in text

Response 30: Yes, NF-water is nuclease free water and we have clarified this in the text.

Line 204:

Next, nuclease-free water (17 μ L) was added to cover the magnetic beads.

Point 31: Line 246 – Did UB6 have stable expression between the groups?

Response 31: Yes, U6 had stable expression between the groups.

Point 32: Line 251 – Change from ‘images’ to ‘Figures were generated’.

Response 32: Thank you for your suggestion, we have changed from ‘images’ to ‘Figures were generated’.

Line 263-264:

The figures were generated using GraphPad Prism 9 (La Jolla, CA, USA).

Point 33: For all figures where appropriate please state what the values presented are e.g. mean \pm SEM

Response 33: Thank you for your reminder, we have added this.

Line 281-282:

The values were presented as mean \pm SEM.

Line 296:

The values were presented as mean \pm SEM.

Line 324:

The values were presented as mean \pm SEM.

Point 34: The n in all figure legends needs clarified. E.g. in figure 1 you have n=10 – is this only 10 ewes total? Or is this 10 ewes that lambd in march and 10 that lambd in august?

Response 34: Thank you for your reminder, we have clarified it.

Line 279-281:

(D) Rectal temperature of pregnant Hu sheep (n = 10, ten Hu sheep in each group). (E) Respiration rate of pregnant Hu sheep (n = 10, ten Hu sheep in each group).

Line 291-292:

Figure 2. Body measurement traits of the normal and IUGR lambs (n = 6, six lambs in each group).

Line 317-318:

Figure 3. Isolation and identification of exosomes from umbilical plasma in normal and IUGR lambs (n = 3, three exosomes samples in each group).

Line 345-346:

Figure 4. Small RNA sequencing results of exosomes derived from umbilical plasma in normal and IUGR lambs (n = 3, three exosomes samples in each group).

Line 362-363:

Figure 5. Differentially expressed miRNAs of exosomes originated from umbilical plasma in CON and HS-IUGR groups (n = 3, three exosomes samples in each group).

Line 399-401:

Figure 7. Expression analysis of significantly expressed miRNAs in exosomes derived from umbilical plasma of normal and IUGR lambs (n = 3, three exosomes samples in each group).

Point 35: The exosome sizes that you present were approx. 140nm whereas in the introduction you stated that exosomes are between 40 – 100nm. Please address this discrepancy and discuss further.

Response 35: Thank you for your reminder, we have changed the description in the introduction.

Line 58-59:

Exosomes, which are membranous vesicles with a diameter of 30-150 nm, are produced in cells and secreted by living cells [14].

Reference:

[14] Théry C, Zitvogel L, Amigorena S. Exosomes: composition, biogenesis and function. *Nat Rev Immunol.* 2002, 2(8): 569-79. <https://doi.org/10.1038/nri855>.

Point 36: How does the 71 miRNAs identified in the umbilical plasma exosomes compare with previous literature in ruminants and humans?

Response 36: Thank you for your reminder, we have added the comparison of the miRNAs identified in the umbilical plasma exosomes compare with previous literature in ruminants and humans.

Line 480-485:

There were 13 significantly expressed miRNAs of exosomes in the umbilical arterial serum compared to the umbilical venous serum of lambs [20]. 116 and 226 miRNAs of umbilical cord blood were differentially expressed in IUGR piglets and IUGR fetuses [45]. The differentially expressed miRNAs were less than the previous studies, the discrepancy may be due to the difference in species.

References:

- [20] Cleys ER, Halleran JL, McWhorter E, Hergenreder J, Enriquez VA, da Silveira JC, et al. Identification of microRNAs in exosomes isolated from serum and umbilical cord blood, as well as placentomes of gestational day 90 pregnant sheep. *Mol Reprod Dev.* 2014, 81(11): 983-93. <https://doi.org/10.1002/mrd.22420>.
- [45] Luo J, Fan Y, Shen L, Niu L, Zhao Y, Jiang D, et al. The pro-angiogenesis of exosomes derived from umbilical cord blood of intrauterine growth restriction pigs was repressed associated with miRNAs. *Int J Biol Sci.* 2018, 14(11): 1426-1436. <https://doi.org/10.7150/ijbs.27029>.

Point 37: What was the litter size of the animals used in this study? Were all animals used singles? Were any of the animals compared littermates as this would significantly decrease the n of your study and the significance?

Response 37: Thank you for your reminder, six Hu male lambs were selected as the experimental lambs in each group, and we used one male lamb of three lambs born to a ewe as an experiment lamb in each group, that is, six Hu male lambs were born in different ewes of each group, so this would not significantly decrease the significance in this study.

Point 38: What was the sex of the lambs used?

Response 38: Thank you for your reminder, the male lambs were used in this study.

Line 109-112:

The umbilical venous blood samples were collected immediately after the Hu lambs were delivered from two groups: six normal male lambs (CON group) born in March, and six IUGR male lambs (HS-IUGR group) born in August.

Point 39: What were the ewes fed?

Response 39: The ewes were fed silage made by the sheep farm.

Point 40: Line 428-430 – This is a bold statement. If you are stating this you need to refer to papers that show the changes in P4 you are referring to here.

Response 40: Thank you for your reminder, we have changed this sentence.

Line 447-448:

This suggests that the content of exosomes in serum changes during pregnancy.

Point 41: Line 478 – I do not agree with this sentence. As the samples were taken at term you do not know if it is ‘dysfunctional cord blood’ that caused the IUGR or if differences in the cord blood are a consequence of IUGR (which is the more likely option)

Response 41: Thank you for your suggestion, we have changed this sentence.

Line 501:

The differences in the umbilical cord blood are a consequence of IUGR.

Response to Reviewer 2 comments:

The authors highlighted that sheep are vulnerable to heat stress when the temperature rises high degrees. Thus, during the summer the ewes that experience high temperatures lead to heat stress. Additionally, when the sheep is pregnant and exposed to heat stress, it can negatively impact the fetal growth and the weight of the lambs. This condition is known as intrauterine growth restriction (IURG). They said that IUGR occurs when the weight of the fetus is less than two standard deviations of the average weight or below the 10% of the normal weight of health animals with the same age. Furthermore, the author raised that small vesicles (exosomes) play a key role during cellular communication. It has already been demonstrated that exosomes can represent physiological or pathological conditions. Thus, exosomes could play a critical role in pregnancy diagnosis, embryo implantation and also during the IURG. In this sense, the authors collected umbilical cord blood from ewes pregnant during the spring (control group – with healthy fetuses) and from ewes during the summer and with IURG fetus. They isolated the exosomes from the plasma and performed RNA sequencing (small RNA). They found that oar-miR-411a-5p was significantly downregulated in exosomes derived from umbilical plasma of IUGR lambs, while oar-miR-200c was significantly upregulated in the HS-IUGR group. When they performed GO and KEGG enrichment analysis observed that the target genes were involved in the Wnt, TGF-beta, and Rap1 signaling pathway. In general, the results are interesting, the article is well written, however the authors should to answer some points highlighted below.

Point 1: The IURG is a condition where animals are born smaller than expected and typically have lower viability and survival rates. Several factors can lead to this condition, as well as, i) under nutrition, heat stress (stated by the authors) and ii) the number and the sex of fetuses per

litter. As reported by the authors, Hu sheep are known for their long estrus cycle and high litter size. Did the authors take these other parameters into account? Please, clarify it.

Response 1: Thank you for your reminder, we have taken these parameters into account. First, the pregnant ewes during the spring and summer were raised with the same feed, so the pregnant ewes were under the same nutrition. Moreover, six Hu male lambs were selected as the experimental lambs in each group, and we used one male lamb of three lambs born to a ewe as an experiment lamb in each group, that is, six Hu male lambs were born in different ewes of each group. Collectively, we have made every effort to ensure that heat stress during pregnancy in ewes led to the IUGR.

Line 105-106:

The pregnant ewes were fed under the same management conditions at Taicang sheep farm, Jiangsu province, China.

Line 109-112:

The umbilical venous blood samples were collected immediately after the Hu lambs were delivered from two groups: six normal male lambs (CON group) born in March, and six IUGR male lambs (HS-IUGR group) born in August.

Point 2: Did the animals are under the same nutritional condition during the spring and summer? If not, how the author guarantee that the IUGR was related to the heat stress and not with nutritional status? How it can interfere in the results?

Response 2: The pregnant ewes during the spring and summer were raised with the same feed, so the pregnant ewes were under the same nutrition.

Line 105-106:

The pregnant ewes were fed under the same management conditions at Taicang sheep farm, Jiangsu province, China.

Point 3: Did the number and the sex of the fetuses are the same for both groups (control vs IUGR)?

Response 3: Yes, the number and the sex of the fetuses are the same for both groups (control vs IUGR). Six Hu male lambs were selected as the experimental lambs in each group, and we used one male lamb of three lambs born to a ewe as an experiment lamb in each group, that is, six Hu male lambs were born in different ewes of each group.

Line 109-112:

The umbilical venous blood samples were collected immediately after the Hu lambs were delivered from two groups: six normal male lambs (CON group) born in March, and six IUGR male lambs (HS-IUGR group) born in August.

Point 4: In the introduction, the authors support very well that pregnant ewes submitted to heat stress can present IUGR. However, we know that it is not just heat stress that causes this condition. Therefore, authors should add a short paragraph in the introduction mentioning that

other factors can also lead to the IUGR condition. In order to keep the introduction not too long, the authors should formulate some paragraphs about heat stress to fit it.

Response 4: Thank you for your suggestion, we have added a short paragraph in the introduction mentioning that other factors can also lead to the IUGR condition.

Line 31-33:

There are many factors that cause IUGR, such as maternal malnutrition, placental dysfunction, and fetal factors, genetic factors also account for a major proportion of the occurrence of IUGR [2, 3].

References:

- [2] Saenger P, Czernichow P, Hughes I, Reiter EO. Small for gestational age: short stature and beyond. *Endocr Rev.* 2007, 28(2): 219-51. <https://doi.org/10.1210/er.2006-0039>.
- [3] Barker DJ, Eriksson JG, Forsén T, Osmond C. Fetal origins of adult disease: strength of effects and biological basis. *Int J Epidemiol.* 2002, 31(6): 1235-9. <https://doi.org/10.1093/ije/31.6.1235>.

Point 5: On page 3, line 63 exclude in cells "...produced in cells and secreted by living cells." change it to "produced and secreted by living cells."

Response 5: Thank you for your suggestion, we have changed "...produced in cells and secreted by living cells." to "produced and secreted by living cells."

Line 58-59:

Exosomes, which are membranous vesicles with a diameter of 30-150 nm, are produced and secreted by living cells [14].

Point 6: The topic Nanoparticle Tracking Analysis in the M&M section should be more detailed. For example, which equipment was used to the analysis? How many records (videos) were made per sample? How many samples were analyzed and were the in duplicates? Etc...

Response 6: Thank you for your reminder, we have added more detail on the topic of Nanoparticle Tracking Analysis in the M&M section.

Line 164-167:

We used a nanoparticle tracking analyzer (PARTICLE METRIX, ZetaVIEW) to detect them. Each sample was imaged in six photos, including the field of view at 1 μ m, 100 nm, 200 nm, and 500 nm. Six samples were analyzed and three duplicates were in each group.

Point 7: Please, indicate in the text the total volume of plasma utilized by the authors to isolate the exosomes.

Response 7: Thank you for your suggestion, we have indicated in the text the total volume of plasma utilized by the authors to isolate the exosomes.

Line 124:

The frozen plasma sample (4 mL) was thawed at 37 °C.

Point 8: On page 12, line 261 there is a mistake in the months of the temperature-humidity index from the summer period. It should be “Jun, Jul and Aug” instead “Jan, Jul and Aug”. Please correct it.

Response 8: We are sorry for this mistake and have corrected it.

Line 273-274:

most pregnant Hu sheep underwent high and even severe stress in Jun, Jul, and Aug (Figure 1C).

Point 9: I do not know if it is only in my file but the quality of the images from the TEM, the NTA graphs and tables in the Figure 3 are not good and should be improved.

Response 9: Thank you for your reminder, we changed the images from the TEM, the NTA graphs and tables in Figure 3.

Point 10: On page 16, Figure 3D the authors used only one exosomal biomarker (CD81). How the authors guarantee there is no contamination from cellular components in the plasma samples? Why the authors did not used a negative marker for exosomes, as well as: GRP78,

Calnexin, cytochrome C or Tomm20? One western blot showing no contamination with cellular components (negative marker for exosomes) should be provided by the authors.

Response 10: Thank you for your suggestion, we have added the Calnexin result of the western blot in exosomes and 293T cells.

Line 308-312:

Calnexin is an endoplasmic reticulum-related protein, which accelerates protein folding and assembly. Compared to 293T cells, there was no Calnexin protein in exosomes. This finding indicated that the separated vesicles were indeed exosomes and free of somatic cell contamination derived from umbilical plasma in both IUGR and normal lambs, as demonstrated in Figure 3D.

Point 11: In the Western Blot what was the protein concentration used in the 293T cell lysate? This is an excess of protein, which makes it difficult to visualize the band correctly.

Response 11: Thank you for your reminder, the protein concentration used in the 293T cell was 2 µg/uL, which unaffected the protein expression of exosomes in the western blot.

Point 12: In the Venny diagram and Heatmap the names of the miRNAs are too small and unreadable.

Response 12: Thank you for your reminder, we have changed the Venny diagram and Heatmap to read the names of the miRNAs more conveniently.

Point 13: On the page 28, lines 475-477 “This alteration could then have led to the modulation of target genes, ultimately disrupting the signaling pathways and impairing the development of skeletal muscle in IUGR lambs.” the authors assume that the miRNAs are altering the genes, however they did not test this hypothesis. In this way, add to the end of the sentence. "However, this hypothesis was not effectively tested in this study."

Response 13: Thank you for your suggestion, we have added this sentence.

Line 495-498:

This alteration could then have led to the modulation of target genes, ultimately disrupting the signaling pathways and impairing the development of skeletal muscle in IUGR lambs. However, this hypothesis was not effectively tested in this study.

Reviewers' comments:

Reviewer #1 (Remarks to the Author):

Thank you for addressing many of the concerns. I do however still have some issues.

Point 1 While the information added in response to this point is important and partially addresses the question, it does not address the well-established fact that there are other differences between March and August lambs which is not accounted for by heat stress. This at least needs to be discussed somewhere in the discussion.

Point 2 If the incidence is approximately 30% there are 70% of the lambs born under heat stressed conditions that are not IUGR. Physiologically you need to have non-IUGR animals as a control as it is well established that IUGR animals have many physiological differences under normal conditions so comparing a 'normal' IUGR with a heat stressed IUGR is not a good comparison and presumably will be masking potential differences. This study needed to have normal animals, normal IUGR animals, heat stressed normal birthweight animals, and heat stressed IUGR.

Point 12 The information provided is not the breeding date, this is the gestation length. My question is what was the variation in date that the ewes were bred e.g. were they all bred the 1st 2 weeks of march, was there a 3 month spread etc. This is important as the data you have presented is all related to the calendar dates but not necessarily day of development. Please also consider the day of gestation into the analyses performed.

Point 38: Why were male lambs selected? Would you expect to see different results in female lambs?

Point 39: please provide this information in the text. How much per day/was it fed ad lib? Were the ewes housed outdoors with access to pasture or indoors? What is the nutritional composition of this silage? Does this meet the nutritional requirements? Was feed intake or body condition of the ewes monitored throughout the study as it is well established that heat stress affects these variables?

Reviewer #2 (Remarks to the Author):

The authors addressed all the points raised in a satisfactory manner. I believe that the quality of the manuscript was improved. In this way, my suggestion is to accept the manuscript for publication.

Response to Reviewers Comments

Dear Reviewers,

Thanks very much for taking your time to review this manuscript. We really appreciate all your generous comments and suggestions! According to your advice, all of your questions were answered one by one. Please find my itemized responses below and my revisions in the re-submitted files.

Response to Reviewer 1 comments:

Thank you for addressing many of the concerns. I do however still have some issues.

Point 1: While the information added in response to this point is important and partially addresses the question, it does not address the well-established fact that there are other differences between March and August lambs which is not accounted for by heat stress. This at least needs to be discussed somewhere in the discussion.

Response 1: There are other differences between March and August lambs, but we do not measure that, more importantly, we only focus on the body measurement traits of lambs to select the normal and IUGR lambs for the collection of umbilical plasma.

Line 440-441:

There were other differences between the normal and IUGR lambs, which might be resulted from genetic factors.

Point 2: If the incidence is approximately 30% there are 70% of the lambs born under heat stressed conditions that are not IUGR. Physiologically you need to have non-IUGR animals as a control as it is well established that IUGR animals have many physiological differences under normal conditions so comparing a 'normal' IUGR with a heat stressed IUGR is not a good comparison and presumably will be masking potential differences. This study needed to have normal animals, normal IUGR animals, heat stressed normal birthweight animals, and heat stressed IUGR.

Response 2: Thank you for your rigorous suggestion, in fact, we compare the differences between the normal and heat stress-induced IUGR lambs, not normal IUGR lambs. In our future studies, we will explore the differences among the normal lambs, normal IUGR lambs, normal lambs under heat stress, and heat-stressed IUGR lambs.

Point 3: The information provided is not the breeding date, this is the gestation length. My question is what was the variation in date that the ewes were bred e.g. were they all bred the 1st 2 weeks of march, was there a 3 month spread etc. This is important as the data you have presented is all related to the calendar dates but not necessarily day of development. Please also consider the day of gestation into the analyses performed.

Response 3: Thank you for your reminder, there were no variations in date between the normal pregnant Hu sheep and heat-stressed pregnant Hu sheep. Specifically, the normal Hu sheep were delivered in the 1st 2 weeks of March, the heat-stressed Hu sheep were delivered in the 1st 2 weeks of August. The pregnancy period was approximately 5 months.

Point 4: Why were male lambs selected? Would you expect to see different results in female lambs?

Response 4: To eliminate the impact of gender on the results, we choose male lambs as the experimental animals.

Point 5: please provide this information in the text. How much per day/was it fed ad lib? Were the ewes housed outdoors with access to pasture or indoors? What is the nutritional composition of this silage? Does this meet the nutritional requirements? Was feed intake or body condition of the ewes monitored throughout the study as it is well established that heat stress affects these variables?

Response 5: Thank you for your suggestion, we have provided this information in the text. This silage meets the nutritional requirements of the pregnant ewes. We do not monitor the feed intake of the ewes throughout the study.

Line 104-108:

The pregnant ewes were fed the silage, and the nutritional composition of the silage was shown in Table S1. The pregnant Hu sheep had *ad libitum* access to feed and water throughout the gestation period. Each pregnant Hu sheep was raised in a single pen. The body condition of the pregnant ewes was observed every day.

Table S1. Nutritional composition of the silage.

Metabolic energy, MJ/kg	10.10
Crude protein, %	11.85
Calcium, %	0.62
Phosphorus, %	0.50
Neutral detergent fibre, %	27.06
Acid detergent fibre, %	11.68